# Assessment of Life Satisfaction of the Physicians of the Silesian Province, Poland

**DOI:** 10.3390/ijerph20065002

**Published:** 2023-03-12

**Authors:** Ewa Niewiadomska, Beata Łabuz-Roszak, Piotr Pawłowski, Klaudia Plinta, Agata Wypych-Ślusarska

**Affiliations:** 1Department of Biostatistics, Faculty of Health Sciences in Bytom, Medical University of Silesia in Katowice, 40-055 Katowice, Poland; 2Department of Neurology, Institute of Medical Sciences, University of Opole, 45-040 Opole, Poland; 3Department of Neurological Rehabilitation, Upper Silesian Rehabilitation Center “Repty”, 42-604 Tarnowskie Góry, Poland; 4Department of Epidemiology, Faculty of Health Sciences in Bytom, Medical University of Silesia in Katowice, 40-055 Katowice, Poland

**Keywords:** life satisfaction, professional satisfaction, anxiety disorders, depressive disorders, physicians

## Abstract

Background: The medical profession is associated with a heavy psychological and physical burden. Specific working conditions can negatively affect the assessment of physicians’ quality of life. The lack of current studies prompted us to evaluate the life satisfaction of the physicians in the Silesian Province in relation to the selected factors (health status, professional preferences, family and material status). Material and methods: The study included 701 physicians and dentists from the Silesian Province aged between 25 and 80 years. It was conducted in 2018 using the Paper and Pencil Interview technique by obtaining non-personalized demographic, anthropometric, socioeconomic, occupational, health and lifestyle data. The following measures were used: the Satisfaction with Life Scale (SWLS), Occupational Satisfaction and the Hospital Anxiety and Depression Scale (HADS). Considering the environmental conditions, the total SWLS scores were analysed in terms of the significance of differences in the groups. Moreover, the SWLS scores underwent multivariate analysis of variance and the correlation analysis of job satisfaction and the presence of anxiety and/or depressive symptoms. Results: Life satisfaction among the physicians and dentists from the Silesian Province was at an average level. Significant predictors included age and economic status. Additionally, significant predictors in the younger subjects (25–50 years) included the Body Mass Index and practising sports. In turn, in the older subjects (50–80 years), these predictors were related to hospital work and being on sick leave. The study found a significant moderate association between life satisfaction and professional satisfaction. Additionally, a significantly lower level of life satisfaction was reported in the subjects who presented with anxiety and/or depressive symptoms. Conclusions: Due to its association with the profession, the mean level of life satisfaction among physicians and dentists prompts verification of crucial spheres related to the physical, emotional, social and material well-being and the activity of the professional group.

## 1. Introduction

The medical profession is of specific nature since it provides help to other human beings. This type of work is associated with a heavy psychological burden. The sense of responsibility for the health and life of patients, taking important decisions (often under time pressure), daily contact with patients who expect improvement in health and recovery, as well as contact with the patient’s family are professional factors that trigger negative emotions that affect health, the quality of work and non-professional life. Furthermore, excessive work and unsatisfactory working conditions that are often reported can negatively affect physicians’ health. In addition, chronic stress and its high level are risk factors for the development of cardiovascular disease and can also lead to mental disorders, such as sleep problems, anxiety symptoms, depression, or addictions. Of note, this professional group is often affected by occupational burnout, which is a state of physical and mental exhaustion from work that occurs as a result of prolonged negative feelings that develop due to workload. This syndrome can occur at any stage of professional development. The above specific working conditions can negatively affect the overall assessment of the quality of life of physicians [1,2,3,4,5].

According to the report on the average life expectancy of physicians (N = 600) prepared by the Silesian Medical Chamber based on the data collected between 2001 and 2006, the male physicians from the Silesian Province lived on average 2.4 years shorter than the national average for men, while Silesian female doctors lived 11.5 years shorter than the national average for women in Poland (Silesian physicians lived to an average of 68.1 years, while female doctors lived to 67.3 years in the study period) [6]. Between 2005 and 2014, the life expectancy of physicians and dentists increased significantly. In 2014, the mean age at death for male and female physicians was 76.0 and 77.9 years, respectively, while in 2005, it was 70.3 and 70.9 years, respectively [6]. The report of the Central Register of Physicians of 5th May 2021 showed that physicians in the pre-retirement age (51–65 years) were the largest group among the physicians working in Poland (33% of the total working physicians) [7]. Of note, more than 15% of active physicians were senior citizens over 65 years of age.

Currently, more attention is paid not only to life expectancy but also to its quality. Among many parameters that are evaluated, the assessment of life satisfaction occupies an important place. El Diener, who developed The Satisfaction with Life Scale (SWLS), explained it as the result of the comparison of the individual’s life situation with the standards adopted by such individuals [8]. Life satisfaction is the individual assessment of one’s life. The more compatible these two aspects are, the higher the satisfaction. Life satisfaction is the result of the balance between experiencing pleasant sensations and a lasting sense of meaning, even in the presence of less pleasurable but fulfiling activities. It can be influenced by health, marriage, family life, relationships with friends, home activities, professional work, the standard of living in a particular country, the place of residence, the amount of leisure time, housing conditions, education, the general standard of living and the adopted values. Additionally, the following are also important: aspirations, the level of realization of dreams and goals, personality traits, genetic predisposition, attitude to life and emotions [5,9,10].

According to the literature, there is a lack of current research on the assessment of life satisfaction among physicians in the Silesian Province, Poland. Furthermore, little data are available on their health status, including mental health, professional preferences, family and material status. Therefore, the above aspects were addressed in this study. The model of the quality of life developed by Fence and Perry was the basis of the analysis. Introduced in 1996, the model includes the following crucial spheres and criteria describing the quality of life: physical well-being (health, physical fitness and safety), emotional well-being (mental status and self-esteem), social well-being (family and social relationships), material well-being (financial status, possession) and development and activity (active participation in life) [11].

So, the aim of the study was to evaluate life satisfaction among Silesian physicians and to analyse its association with demographic, anthropometric and socioeconomic data, moreover with professional satisfaction, work environment, lifestyle and health status, including depressive and anxiety symptoms.

## 2. Material and Methods

### 2.1. Participants

The Silesian Province is the second largest province in Poland, with a population of about 4.6 million inhabitants, which accounts for 12% of the country’s total population. An extensive network of healthcare units is the response to the high population density in this region. According to the 2019 Central Register of Physicians, the population of professionally active physicians and dentists was 20,828 (including 4065 dentists) in the Silesian Province, Poland [7]. According to the estimation of the expected sample size (confidence interval 95%, acceptable error 5%), the minimum sample size was set at 378. Adopting the margin of uncertainty, we distributed 1000 questionnaires among physicians and dentists with a return rate of 70.1%. Dentists constituted less than 1% (n = 6); therefore, they were not distinguished as a separate group.

Given the age structure of the physicians in which the median age was 50 years according to the data collected from the Supreme Medical Chamber (per 2017) and the Silesian Medical Chamber (per 2019), the study group was divided into two age categories, i.e., 25–50 years (25 ≤ X < 50) and 50–80 years (50 ≤ X ≤ 80) [12]. To maintain representativeness, the demographic structure of the group was considered. According to the data from 2017, women accounted for 57.7% of all physicians in Poland, while men accounted for 42.3% [12]. The selection of the study group was determined by the criterion of no significant differences in the structure by gender compared to the population structure. According to the register of the Supreme Medical Chambers of 2019 on medical and dental specialties, the most common ones in the Polish population were distinguished [7]. The professional groups were compatible with the Polish population of physicians, depending on the specialty.

The inclusion criteria were as follows: the complete or limited right to practice as a medical doctor and/or dentist and the membership in the Medical Chambers of the Silesian Province (Częstochowa, Silesia, and Bielsko; Poland). Physicians and dentists who were not professionally active and/or those who were not members of the Medical Chambers of the Silesian Province were excluded.

### 2.2. Methods

The study was conducted from January to December 2018 using a paper questionnaire (Paper and Pencil Interview—PAPI method) in the form of an anonymous survey containing closed and open questions (standard information on sociodemographic, economic, occupational, lifestyle and health data).

Life satisfaction was assessed according to SWLS scale (Satisfaction with Life Scale) [13] and the presence of anxiety and/or depressive symptoms according to the Hospital Anxiety and Depression Scale (HADS) [14]. Moreover, the level of professional satisfaction was evaluated [15].

SWLS contains five items (e.g., “I am satisfied with my life”, “I have achieved the most important things I wanted in my life”). Using a 7-point Likert scale, the respondent assesses to what extent each of the statements applies to his/her life. The total number of points ranged from 5 to 35 points [13]. The SWLS scale was obtained from the Laboratory of Psychological Tests of the Polish Psychological Association.

HADS is a self-report questionnaire. Patients are asked to choose one answer out of four possibilities for each question. Anxiety questions are marked as “A” (7 questions) and depression questions as “D” (7 questions). They are arranged alternately. The scores (from 0 to 3 points) for each question for “A” and separately for “D” are added together to obtain two scores: depression (D) and anxiety (A). A total score of 0 to 7 points indicates no disorder, values from 8 to 10 points are borderline and ≥11 points indicate the presence of anxiety (A) and/or depressive symptoms (D) [14].

Professional satisfaction was measured according to the Canadian questionnaire validated for Polish conditions by Pena-Sanchez et al. [15]. The questionnaire consists of 17 questions evaluating various aspects of the doctor’s work (e.g., relationship with other doctors, doctor–patient relationship, income, professional development opportunities and opportunities to separate professional duties from personal life). It uses a 6-point scale (very dissatisfied, dissatisfied, rather dissatisfied, somewhat satisfied, satisfied, very satisfied). The total number of points ranged from 17 to 102 points.

### 2.3. Data Analysis

Statistical calculations were made using MS Excel 2019 (Microsoft Corporation, Warsaw, Poland) and STATISTICA 13 (Stat Soft Poland, Cracow, Poland). The overall life satisfaction, job satisfaction and the presence of anxiety and/or depressive symptoms according to the HADS were determined as the total number of points obtained from a particular questionnaire. For the HADS, the following levels were considered: normal <0,7>, borderline <8,11> and anxiety/depressive symptoms <12,21>. Measurable data were characterized using mean (X) and standard deviation (SD), and median and quartile range (QR). The percentage was used for the nominal data. The Shapiro–Wilk test was applied to verify the conformity of the variables to normal distribution.

The significance of differences in the means was tested using the Student’s t-test for two groups and the ANOVA test for three or more groups. The conformity of distributions in the groups was tested using the Mann–Whitney U and Kruskal–Wallis tests for skewed distributions. Multiple comparisons were performed using post-hoc tests for ANOVA analysis of variance and the Kruskal–Wallis test. The dependence of variables was assessed by determining Spearman’s r correlation coefficient with the corresponding significance test. ANOVA analysis of variance was used in the multivariate analysis with consideration given to the predictors that had a significant effect on the dependent variable in the univariate models. Additionally, variables whose strength of correlation with the others was not significant were considered. A statistical significance level of *p* < 0.05 was used as a criterion for including a variable in the model.

## 3. Results

A total of 701 physicians and dentists were surveyed, including 336 (47.9%) women and 365 (52.1%) men aged 25–80 years. The study results were presented in two age groups, i.e., 25–50 years <25,50) (*n* = 492; 70.2%) and 50–80 years <50,80> (*n* = 209; 29.8%). According to the SWLS, life satisfaction was assessed based on five statements and the level (from 1 to 7) to which each statement related to life. The overall level of life satisfaction of the physicians was 19 ± 4.2 points (range 6–35 points). The respondents tended to disagree with the high scores of their present quality of life. However, a neutral response was predominant when satisfaction with life and achievements were assessed. Complete life satisfaction (35 points) was expressed by two physicians (0.3%). Figure 1 shows the detailed results of the assessment of life satisfaction in the group of physicians.

There was a significant effect of the subjects’ age on life satisfaction. A significantly higher level of life satisfaction was associated with younger physicians aged 25–50 years compared to the older group aged 50–80 years (19.3 ± 4 vs. 18.2 ± 4.6; *p* < 0.001). Therefore, the analyses were performed for two separate age groups.

Life satisfaction was significantly higher in younger male doctors than in female ones (*p* = 0.002). In the older age group, there was no difference between men and women. Additionally, a significantly higher level of life satisfaction was found in the subjects with low or normal BMI, in a partnership or marriage, and in those who declared a good economic status. Having children had no effect on life satisfaction (Table 1).

A significantly higher level of life satisfaction was found among the physicians who were not specialists, worked in a hospital, had shorter work experience, had longer holiday leave and enjoyed recreational trips more often (Table 2).

In both age groups, a significantly lower level of life satisfaction was found in those with chronic diseases and those requiring pharmacotherapy. In turn, a significantly higher level of life satisfaction was found among the physicians who had never been on sick leave, practiced sport and performed regular physical activity, had never followed any diet and did not report chronic fatigue. Smoking and hours of sleep did not influence life satisfaction (Table 3).

The results of a significant model of the multivariate analysis of variance showed a significantly higher level of life satisfaction in the case of good assessment of economic status and among younger physicians (Figure 2).

Good economic status was significantly important for a higher quality of life in both age groups (Figure 2).

In younger physicians, low BMI and practicing sports had a significant additional influence, while in older physicians, working in the hospital and not taking sick leave were significant (Figure 2).

The overall level of professional satisfaction of the doctors was 69.4 ± 8.5 points (range 39–99 points; scale range 17–102 points). The analysis of the relationship between professional satisfaction and life satisfaction indicated a moderate, significant association between these two aspects (R = 0.34; *p* < 0.0001) for the overall study group and the physicians aged 25–50 (R = 0.28; *p* < 0.0001) and those aged 50–80 years (R = 0.42; *p* < 0.0001). However, the strength of the relationship for those aged 50–80 years was almost twice as high compared to the younger group.

In addition, the level of life satisfaction was significantly lower in those who presented with anxiety and/or depressive symptoms for the study population and individual age groups (Table 4).

## 4. Discussion

The results reflect life satisfaction among the physicians and dentists in the Silesian Province, Poland, in relation to the selected demographic, medical, psychological, socioeconomic and environmental factors. As shown in the study, life satisfaction among the physicians from the Silesian Province, Poland, was at an average level. Age and economic status were significant predictors. Additionally, significant predictors included BMI and practicing sports in the younger subjects (25–50 years). In turn, in the older subjects (50–80 years), these predictors were related to hospital work and being on sick leave. The study found a moderate association between life satisfaction and professional satisfaction. Additionally, lower levels were reported in the subjects who presented with anxiety and/or depressive symptoms.

Another survey on the quality of life in Poland conducted in 2017 by Statistics Poland showed that more than 80% of the respondents reported general satisfaction with life, about 70% declared optimism, and almost 80% had a sense of meaning in life. Only 4.5% of the respondents experienced symptoms of poor well-being, such as nervousness, depression, sadness, exhaustion and fatigue [16]. In turn, an earlier study (2013) linked a high level of life satisfaction with higher education and working in senior management and specialized positions [17]. Of note, according to the Classification of Occupations and Specialties, physicians, nurses and midwives are included in the group of specialists [18]. A marked difference was found in the perception of the quality of life due to material status, i.e., low economic status resulted in a considerably lower assessment compared to those with the highest income [17].

On the other hand, the results of our study indicate an average level of life satisfaction of 19 ± 4.2 points (range 6–35 points) and a significant relationship between it and economic status. The lowest life satisfaction was found among the physicians who assessed their economic status as poor. In addition, the level of life satisfaction was significantly higher among those who were underweight, in a partner relation, without a medical specialty, who worked in the hospital, who worked for less time, took holiday leave more frequently, did not present with chronic diseases or were not on drug therapy, were not on sick leave, practised sport, were involved in regular physical activity, were not on a diet and were well-rested. A moderate association between life satisfaction and professional satisfaction was also found. Additionally, lower levels were reported among the physicians with anxiety and/or depressive symptoms.

As regards life satisfaction, only 27.5% of our respondents indicated that they were satisfied to varying degrees. However, more than 50% of the physicians could not specify it. Furthermore, 60.2% of them disagreed with the opinion that their lives were close to the ideal. This result seems all the more surprising given the 2019 survey on the prestige of professions that was conducted by the Centre for Public Opinion Research [19]. The medical profession was held in high esteem by 80% of the surveyed Poles. According to public opinion, it is one of the most useful professions related to providing help. Therefore, it might seem that personal satisfaction would also be high. The survey was not limited to the analysis of life satisfaction in the professional sphere. However, it should be borne in mind that this aspect also influences overall life satisfaction.

The level of satisfaction with various aspects of life was also assessed by a survey conducted by Statistics Poland in 2013 [17]. While the overall satisfaction in Polish society was rather satisfactory, some disparities were found in certain spheres. The highest levels of satisfaction were related to relationships with other people, family situations, the amount of leisure time, professional situations and leisure activities. In contrast, a high percentage of dissatisfaction was associated with the financial situation, education and health. Multivariate analysis showed which of the above aspects influenced overall life satisfaction, the most significant being the family situation and the health status. The amount of leisure time was not significant as opposed to how this time was spent. There was also a significant positive effect on material living conditions, social relationships, financial situations, education and work situations. Of note, the results of the cause and effect relationship showed that higher life satisfaction was found in women compared to men, people in the second half of life, those in a formal or informal relationship, with children, individuals in good or very good health and those who declared stable life pattern with no breakdown compared to an unstable life pattern, when the occurrence of events clearly improves living conditions, and the residents of the Silesian Province [17].

In turn [17], the study conducted by Statistics Poland revealed that a significantly reduced life satisfaction was found in those with incomplete higher education compared to those with secondary education, unemployed physicians, those in poor or very poor health, poor mental well-being, in social isolation, those who declared an unstable life pattern compared to a stable life pattern, those in the presence of markedly worsening life conditions, in those with a mental breakdown and in those living in poverty [17].

Another cross-sectional epidemiological survey of economically active residents of the Silesian Province aged 45–60 [20] showed average levels of satisfaction in the following spheres: somatic (including health, performance in work and daily life, quality of sleep and rest), social (relationships with people), environmental (security, financial situation, the opportunity to pursue interests, social conditions) and psychological (self-satisfaction in daily life, the frequency of experiencing negative feelings such as anxiety and depression). The highest levels were found for the social and psychological domains of life, while the lowest levels were found for the somatic domain. A significant effect of the marital situation was observed in relation to health status, i.e., good health was observed more often in married people compared to single individuals. Moreover, a higher education level and intellectual work determined better health status. These factors significantly affected the psychological, social and environmental spheres of life. Kowalska et al. (2011) indicated a strong need for the formation of health-promoting behaviours of employees as part of the training provided by the services responsible for supervising occupational health and safety [20].

In the review of studies on occupational hygiene, Żołnierczyk confirmed the strong negative impact of long working hours on the overall quality of life [21]. That study showed the occurrence of work–family conflict as a basis for the development of depressive behaviour. Evidence also showed that long-term work in a harmful environment had a significant impact on lower life satisfaction, which was more evident among older individuals and women. This is a serious problem in the medical community, considering the results of this study and the significant percentage of physicians surveyed (42.5%) who were employed in more than one medical facility. The analysis also showed that much more than half of the physicians reported the occurrence of chronic fatigue.

Given the high rate of feminization in the medical profession, the quality of life of women in Polish society was analysed by Kowalska et al. Their study assessed job satisfaction and the quality of life of women aged 45–60 in the Silesian Province. Their study showed a higher quality of life for professionally active women compared to non-working women [22]. At the same time, half of the professionally active women confirmed the occurrence of anxiety related to the work environment. Of note, outside of their professional duties, much more than half of the women surveyed provided care for children, grandchildren or elderly parents. The assessment of physical and mental condition at the average level of 74% and 59.5% of the maximum score was rather related to the good health of the respondents. However, better health was reported in women who worked under stress-free conditions. A higher quality of life was reported by women who were better educated, professionally active and satisfied with their workplace. However, we found no significant effect of the place of residence, age or additional family responsibilities on the quality of life.

Peplińska et al. (2017) also confirmed the significant importance of professional work in shaping a sense of purpose and meaning in life [23]. In particular, attention was paid to occupational commitment, occupational status and experiences resulting from work activity. It was emphasized that occupational satisfaction allowed the individual to see opportunities for self-realization and to derive satisfaction from the performed function. It also reduced negative feelings resulting from fulfiling social roles. On the other hand, their study found a negative impact of stress and problems related to the work environment on the quality of family relationships and personal life, which negatively affected the meaning of life.

Given the results indicating the determinants of life satisfaction of the surveyed doctors, it would be appropriate to consider whether it depends on occupational burnout. The latter was not the subject of the study; hence, it is not possible to draw such conclusions. However, a higher level of life satisfaction among physicians without specialty and those with shorter seniority could suggest such a relationship. Occupational burnout is a psychological syndrome related to emotional exhaustion (lack of energy, enthusiasm for work), depersonalization (indifference to work) and a sense of professional ineffectiveness, manifested by a reduced sense of achievement at work [24]. A study conducted among Chinese medical students showed that occupational burnout (defined as academic burnout) significantly affected life satisfaction [25].

Early detection of professional burnout is important, and for this purpose, the Maslach Burnout inventory should be used. Research conducted among clinicians reveals that the self-assessment of occupational burnout omits half of the real cases [24,26].

In addition, special attention should be paid to the mental health of doctors working in hospitals, who are particularly vulnerable due to workplace-specific factors [27]. On the other hand, it is worth mentioning the elements that improve the quality of life and work. Kase and Doolittle, in their work, indicated the following: mental resilience, social connections (friendships, in and outside the workspace), expanded horizons (hobbies and activities outside of medicine and work), personality type (as having both introverted and extroverted features), balanced perspective (the importance of equanimity, maintaining a balanced perspective of the challenges and successes of their role) [5]. Howe, in turn, emphasizes the importance of education to enhance mental resilience in the medical profession (through simulations, role-playing, discussion of cases and ethical dilemmas) to build a better quality of life [28].

In view of the average level of life satisfaction of the physicians and its connection with the profession, the results prompt verification of crucial spheres related to the physical, psychosocial and material well-being and the activity of this professional group. In studies on life satisfaction among physicians and other medical personnel, the aspects of occupational burnout and psychological determinants of these processes should also be considered. For instance, the impact of perfectionism (especially maladaptive perfectionism) on the indicators of life satisfaction should be considered. Severe self-criticism when goals are not achieved is associated with many psychological dysfunctions, including depression. Maladaptive perfectionism, occupational burnout and a low level of life satisfaction are strongly correlated [25]. Therefore, medical institutions should take early measures to prevent these processes, which can negatively affect the quality of physicians’ work, and, thus, the quality of medical care given to patients.

## 5. Conclusions

Due to its association with the profession, the mean level of life satisfaction among physicians and dentists prompts verification of crucial spheres related to the physical, emotional, social and economic well-being and the activity of this professional group.

## 6. Limitations

Our study has some limitations. Firstly, it was a cross-sectional study. Secondly, the physicians and the dentists were not analysed separately because only a few dentists (n = 6) returned the questionnaire, and they ultimately accounted for less than 1% of all respondents. Thirdly, we do not analyse occupational burnout and its effect on life satisfaction in Silesian doctors. In the future, we plan to extend the research to include also this parameter.

## Figures and Tables

**Figure 1 ijerph-20-05002-f001:**
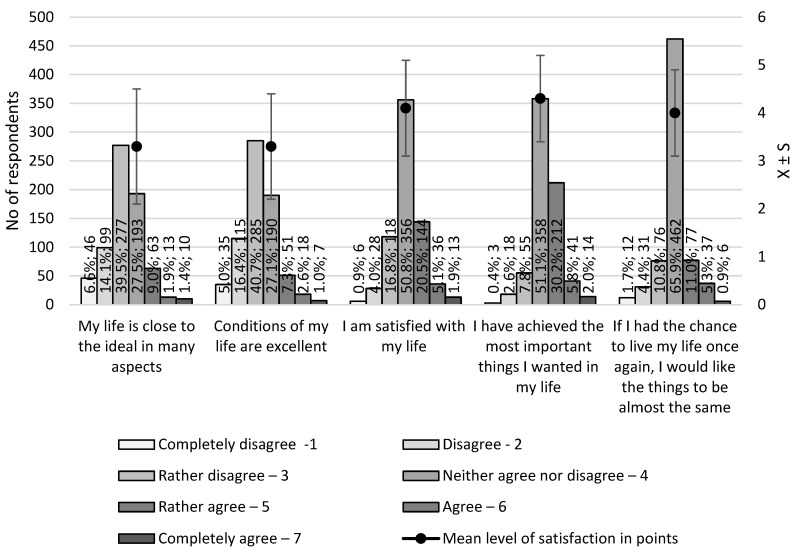
Results of the Satisfaction with Life Scale (SWLS).

**Figure 2 ijerph-20-05002-f002:**
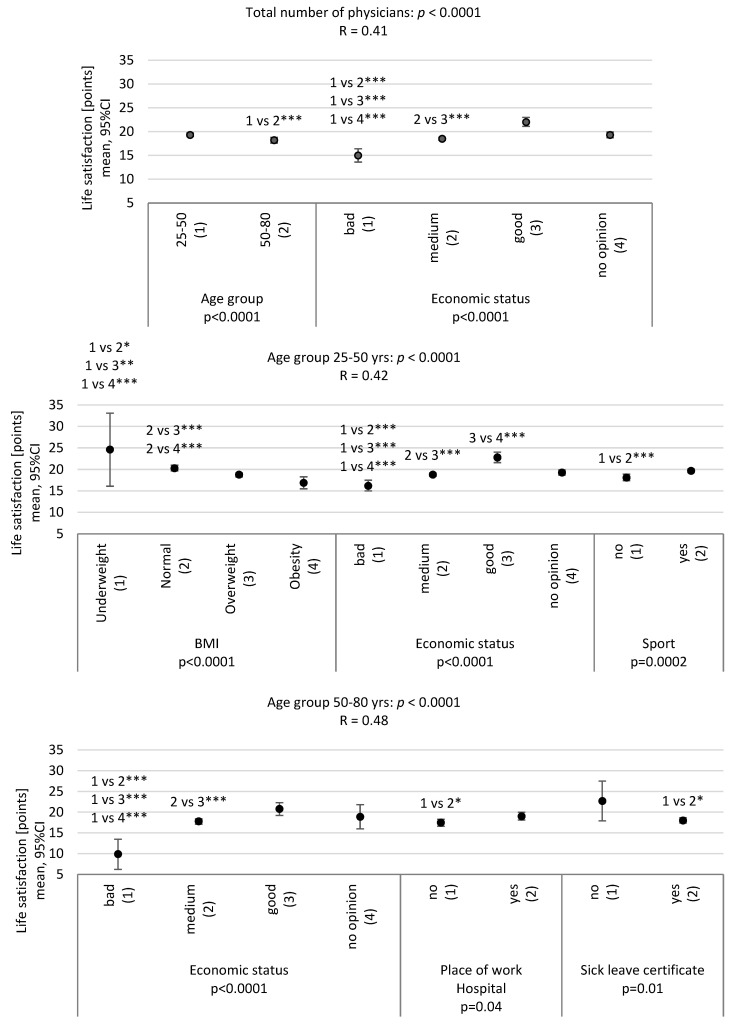
Analysis of variance model for the dependent variable (life satisfaction) for the study group (N = 701) and in accordance with the age group (25–50 yrs; 50–80 yrs). The result of the analysis of variance as a box-and-whisker plot for X, 95% CI—the mean and 95% Confidence Interval for the mean; *p*-value—the result of the multivariate ANOVA for quantitative characteristics with univariate significance tests; results of multiple comparisons: * *p* < 0.05; ** *p* < 0.01; *** *p* < 0.001.

**Table 1 ijerph-20-05002-t001:** Influence of demographic, anthropometric and socioeconomic data on the level of life satisfaction among physicians.

Demographic and Anthropometric Data		Total 701 (100%)		Age Groups			
				[25,50) yrs 492 (70.2%)		[50,80) yrs 209 (29.8%)	
		Life Satisfaction X ± S M(QR)	*p*-Value	Life Satisfaction X ± S M(QR)	*p*-Value	Life Satisfaction X ± S M(QR)	*p*-Value
Gender	women	18.9 ± 4.3 18(3)	0.140	18.9 ± 4.2 19(3)	0.002	18.8 ± 4.5 18(4)	0.110
	men	19.1 ± 4.2 19(4)		19.8 ± 3.8 20(3)		17.6 ± 4.6 18(5)	
BMI	underweight (1)	24.6 ± 6.9 23(9)	<0.0001 1 vs. 4 * 2 vs. 3 * 2 vs. 4 *** 3 vs. 4 *	24.6 ± 6.9 23(9)	<0.0001 1 vs. 4 * 2 vs. 3 ** 2 vs. 4 **	-	0.230
	normal (2)	19.8 ± 4.5 19(4.5)		20.3 ± 4.2 20(4)		17.4 ± 5.2 18(6)	
	overweight (3)	18.7 ± 4 18(3)		18.8 ± 3.6 19(3)		18.5 ± 4.6 18(5)	
	obesity (4)	17.1 ± 3.1 18(4)		16.9 ± 3.1 18(4)		17.3 ± 3.1 17(4)	
Marital status	single (1)	18.7 ± 4.2 18(3)	0.006 1 vs. 2 * 2 vs. 4 *	18.8 ± 4.2 18(3)	0.009 1 vs. 2 *	15.9 ± 3.4 16(5)	0.290
	in a partner relationship (2)	20.2 ± 3.8 20(4)		20.2 ± 3.7 20(4)		20.1 ± 4.3 21(6)	
	married (3)	19.2 ± 4.3 19(4)		19.6 ± 4 19(3)		18.5 ± 4.6 18(5)	
	divorced (4)	17.6 ± 3.5 18(3)		18.6 ± 2.2 19(3)		16.6 ± 4.3 18(5)	
	widow/ widower (5)	17.6 ± 4.6 18(6)		-		17.6 ± 4.6 18(6)	
Children	no	19.2 ± 4.3 19(3)	0.860	19.3 ± 4.3 19(3)	0.360	16.9 ± 2.8 18(2)	0.230
	yes	18.9 ± 4.2 19(4)		19.4 ± 3.6 19(3)		18.3 ± 4.7 18(5)	
Subjective assessment of the economic status	bad (1)	15 ± 4.2 15(6)	<0.0001 1 vs. 2 *** 1 vs. 3 *** 1 vs. 4 *** 2 vs. 3 *** 3 vs. 4 **	16.2 ± 3.3 15(5)	<0.0001 1 vs. 2 ** 1 vs. 3 *** 1 vs. 4 *** 2 vs. 3 *** 3 vs. 4 ***	9.9 ± 3.9 9(4)	<0.0001 1 vs. 2 ** 1 vs. 3 *** 1 vs. 4 * 2 vs. 3 **
	medium (2)	18.5 ± 3.6 18(3)		18.8 ± 3.5 19(3)		17.8 ± 3.9 18(5)	
	good (3)	22 ± 5.3 21(6)		22.8 ± 5.3 21(5)		20.8 ± 5.2 20(6)	
	no opinion (4)	19.3 ± 2.6 19(2.5)		19.3 ± 2.6 19(3)		18.9 ± 3.1 18(3)	

X ± S—mean and standard deviation; M (QR)—median and quartile range; *p*-value—significance result of the Student’s t-test/Mann–Whitney U test for quantitative characteristics in 2 groups; significance result of ANOVA/Kruskal–Wallis test for quantitative characteristics in 3 or more groups; results of multiple comparisons between groups (1)–(5): * *p* < 0.05; ** *p* < 0.01; *** *p* < 0.001.

**Table 2 ijerph-20-05002-t002:** Influence of data related to professional work environment on the life satisfaction of the physicians.

Professional Work		Total 701 (100%)		Age Groups			
				[25,50) yrs 492 (70.2%)		[50,80) yrs 209 (29.8%)	
		Life Satisfaction X ± S M(QR)	*p*-Value	Life Satisfaction X ± S M(QR)	*p*-Value	Life Satisfaction X ± S M(QR)	*p*-Value
Being a specialist	no	20.4 ± 4.8 20(5)	0.0008	20.4 ± 4.8 20(5)	0.010	-	-
	yes	18.7 ± 4.1 18(3)		19 ± 3.7 19(3)		18.2 ± 4.6 18(5)	
Working at the hospital	no	18 ± 4.3 18(5)	0.0005	19.1 ± 3.5 19(4)	0.620	17.5 ± 4.5 18(5)	0.020
	yes	19.3 ± 4.1 19(4)		19.4 ± 4.1 19(4)		19 ± 4.5 18(4)	
Spending time after duty at the primary workplace	resting	19.3 ± 4.1 19(4)	0.600	19.6 ± 4.2 20(3)	0.100	17.9 ± 3.2 18(3)	0.040
	working at another workplace	19.3 ± 4 19(4)		19.2 ± 3.7 19(3)		19.8 ± 4.5 19(4)	
		R	*p*-Value	R	*p*-Value	R	*p*-Value
Length of professional career (yrs)		−0.15	<0.0001	−0.1	0.020	0.03	0.670
Number of working hours (hours/week)		0.05	0.220	0.05	0.230	−0.02	0.730
Holiday leave (days)		0.1	0.008	0.06	0.140	0.16	0.020
Recreational trips (n/yr)		0.2	<0.0001	0.14	0.020	0.28	<0.0001

X ± S—mean and standard deviation; M (QR)—median and quartile range; *p*-value—significance result of the Student’s *t*-test/Mann–Whitney U test for quantitative characteristics in 2 groups; significance result of ANOVA/Kruskal–Wallis test for quantitative characteristics in 3 or more groups; R—Spearman’s correlation coefficient; *p*-value—test significance result for R (correlation coefficient).

**Table 3 ijerph-20-05002-t003:** Effect of data related to health and lifestyle on life satisfaction of the physicians.

Lifestyle		Total 701 (100%)		Age Groups			
				[25,50) yrs 492 (72.2%)		[50,80) yrs 209 (29.8%)	
		Life Satisfaction X ± S M(QR)	*p*-Value	Life Satisfaction X ± S M(QR)	*p*-Value	Life Satisfaction X ± S M(QR)	*p*-Value
Chronic disease	no	20.5 ± 4.6 20(5)	<0.0001	20.4 ± 4.5 20(5)	<0.0001	22.8 ± 5.6 25(10)	0.004
	yes	18.3 ± 3.8 18(4)		18.5 ± 3.4 18(3)		17.9 ± 4.4 18(5)	
Pharmacotherapy	no	20.6 ± 4.6 20(5)	<0.0001	20.5 ± 4.6 20(5)	<0.0001	21.5 ± 5.3 22(9)	0.010
	yes	18.2 ± 3.8 18(4)		18.4 ± 3.2 18(3)		17.9 ± 4.4 18(5)	
Sick leave certificate	no	20.6 ± 4.5 20(5)	<0.0001	20.5 ± 4.4 20(5)	0.002	22.7 ± 5.2 23(10)	0.030
	yes	18.7 ± 4.1 18(3)		19 ± 3.8 19(3)		18 ± 4.5 18(5)	
Practicing sport	no	17.6 ± 4.2 18(5)	<0.0001	18.1 ± 3.7 18(4)	0.0002	17.1 ± 4.6 17(6)	0.0003
	yes	19.6 ± 4.1 19(4)		19.7 ± 4 19(3)		19.2 ± 4.4 19(3.5)	
Physical activity	irregular	18.9 ± 3.5 19(3)	<0.0001	18.9 ± 3.4 19(3)	<0.0001	18.7 ± 4 19(3)	0.050
	regular	23 ± 5 23(5)		23.2 ± 4.9 23.5(5)		22.1 ± 5.8 21(11)	
Diet	never	19.3 ± 4.4 19(4)	0.001	19.3 ± 4.1 19(4)	0.290	18.9 ± 5.3 18(6)	0.040
	yes at any time	18.3 ± 3.8 18(3)		19.2 ± 3.7 18(2)		17.4 ± 3.6 18(5)	
Smoking	no	19.1 ± 4.2 19(4)	0.150	19.4 ± 4.1 19(4)	0.180	18.3 ± 4.2 18(4)	0.380
	yes	18.5 ± 4.4 18(4)		18.8 ± 3.4 18(3)		17.7 ± 6.2 18(6)	
Alcohol consumption	no	18.8 ± 4.2 18(3)	0.150	18.9 ± 3.8 18(3)	0.010	18.6 ± 4.9 18(6)	0.440
	yes	19.1 ± 4.3 19(4)		19.7 ± 4.1 20(5)		17.9 ± 4.3 18(4)	
Chronic fatigue	yes	17.9 ± 4.7 18(5)	<0.0001	19 ± 4.8 18(5)	0.230	16.9 ± 4.3 17(5)	<0.0001
	no	19.6 ± 3.8 19(3)		19.4 ± 3.7 19(3)		20.2 ± 4.3 19(4)	
Lifestyle		R	*p*-value	R	*p*-value	R	*p*-value
Sleep (hours)		0.010	0.770	−0.05	0.240	0.050	0.430
Pack-years		−0.080	0.360	0.13	0.220	−0.190	0.250

X ± S—mean and standard deviation; M (QR)—median and quartile range; *p*-value—significance result of the Student’s t-test/Mann–Whitney U test for quantitative characteristics in 2 groups; significance result of ANOVA/Kruskal–Wallis test for quantitative characteristics in 3 or more groups; R—Spearman’s r correlation coefficient; *p*-value—test significance result for R (correlation coefficient).

**Table 4 ijerph-20-05002-t004:** Level of satisfaction with life according to the co-occurrence of anxiety and/or depressive symptoms for the total study group (N = 701) and in accordance with the age group.

Age Groups	Life Satisfaction (Points) X ± S M (QR)			*p*-Value	R *p*-Value
	HADS-A [0 ÷ 21]				
	Normal Range 351 (50.1%)	Borderline Result 169 (24.1%)	Anxiety Symptoms 181 (25.8%)		
Total	20.6 ± 4.4 20 (5)	18.3 ± 2.9 18 (3)	16.5 ± 3.5 17 (5)	<0.0001 1 vs. 2 *** 1 vs. 3 *** 2 vs. 3 ***	−0.420 < 0.0001
25–50 yrs	20.4 ± 4.2 20 (4)	18.3 ± 2.6 18 (3)	16.7 ± 3 17 (5)	<0.0001 1 vs. 2 *** 1 vs. 3 *** 2 vs. 3 ***	−0.380 < 0.0001
50–80 yrs	21.8 ± 5.1 21 (8.5)	18.2 ± 3.4 18 (3)	16.3 ± 3.8 17 (5)	<0.0001 1 vs. 2 * 1 vs. 3 *** 2 vs. 3 **	−0.410 < 0.0001
**Age groups**	**HADS-D [0÷21]**			***p*-Value**	**R** ***p*-value**
	**Normal range** **382 (54.5)**	**Borderline result** **184 (26.2)**	**Depressive symptoms ** **135 (19.3)**		
Total	20.6 ± 4.3 20 (5)	17.4 ± 2.9 18 (3.5)	16.6 ± 3.5 17 (5)	<0.0001 1 vs. 2 *** 1 vs. 3 ***	−0.420 < 0.0001
25–50 yrs	20.4 ± 4.1 20 (4)	17.4 ± 2.9 18 (3)	17.2 ± 2.8 18 (4)	<0.0001 1 vs. 3 *** 2 vs. 3 ***	−0.390 < 0.0001
50–80 yrs	21.5 ± 5 20 (7)	17.4 ± 3 18 (4)	16.2 ± 3.8 17 (5)	<0.0001 1 vs. 3 *** 2 vs. 3 ***	−0.420 < 0.0001

1—normative [0, 7]; 2—borderline [8, 11]; 3—anxiety/depressive symptoms [12, 21]; N (%)—number and percentage; X ± S—mean and standard deviation; M (QR)—median and quartile range; *p*-value—significance result of the X2 test/Fisher’s test for qualitative characteristics; significance result of ANOVA/Kruskal–Wallis test for quantitative characteristics in 3 or more groups; results of multiple comparisons: * *p* < 0.05; ** *p* < 0.01; *** *p* < 0.001; R—Spearman’s correlation coefficient; *p*-value—test significance result for R (correlation coefficient).

## Data Availability

The data presented in this study are available on request from the corresponding author. The data are not publicly available due to privacy reasons.

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
