# Peer review of "Assessment of Life Satisfaction of the Physicians of the Silesian Province, Poland"

_ijerph, 2023, doi:10.3390/ijerph20065002_

Round 1
Reviewer 1 Report
I would like to highly recommend somehow implementing the Maslach Burn-out inventory into the survey, or at least the phenomena itself, the burn-out which is quite an important factor could be behind life satisfaction and depression. Even the Burnout phenomenon contains depression as one of its dimensions, so it can be considered as a level or stage of burnout. Besides the references do not seem to cover all the factors of the topic and more international references should be added. As a more quantitative remark or comment is referred to on page 8, where there is no explanation of the data at all but a Table itself the only information on this page. The same problem occurs on pages 4-7, but not in an extensive way.
Author Response
Thank you very much for detailed review of our manuscript and all the comments. Our replies and changes in the manuscript are written in a red.
1.
“I would like to highly recommend somehow implementing the Maslach Burnout inventory into the survey, or at least the phenomena itself, the burn-out which is quite an important factor could be behind life satisfaction and depression. Even the Burnout phenomenon contains depression as one of its dimensions, so it can be considered as a level or stage of burnout. “
We added in the end of the discussion:
Early detection of professional burnout is important, and for this purpose the Maslach Burnout inventory should be used. Research conducted among clinicians reveals that the self-assessment of occupational burnout omits half of real cases [19].
We also added in the Limitations:
Thirdly, we do not analyze occupational burnout and its effect on life satisfaction in Silesian doctors. In the future, we plan to extend the research to include also this parameter.
2.
“Besides the references do not seem to cover all the factors of the topic and more international references should be added. “
We added more international references:
- Bradfield, O.; Jenkins, K.; Spittal, M.; Bismark, M. Australian and New Zealand doctors' experiences of disciplinary notifications, investigations, proceedings and interventions relating to alleged mental health impairment: a qualitative analysis of interviews. Int J Law Psychiatry 2023, 86:101857. doi: 10.1016/j.ijlp.2022.101857.
- Lele, K.; McLean, L.M.; Peisah, C. Beyond burnout I: Doctors health services and unmet need. Australas Psychiatry 2023:10398562231159977. doi: 10.1177/10398562231159977.
- Nguyen, O.T.; Turner, K.; Parekh, A.; Alishahi Tabriz, A.; Hanna, K.; Merlo, L.J.; Hong, Y.R. Merit-based incentive payment system participation and after-hours documentation among US office-based physicians: Findings from the 2021 National Electronic Health Records Survey. J Eval Clin Pract 2023, 29(2), 397-402. doi: 10.1111/jep.13796.
- Kartsonaki, M.G.; Georgopoulos, D.; Kondili, E.; Nieri, A.S.; Alevizaki, A.; Nyktari, V.; Papaioannou, A. Prevalence and factors associated with compassion fatigue, compassion satisfaction, burnout in health professionals. Nurs Crit Care 2023, 28(2), 225-235. doi: 10.1111/nicc.12769.
- Kase, J.; Doolittle, B. Job and life satisfaction among emergency physicians: A qualitative study. PLoS One 2023, 18(2):e0279425. doi: 10.1371/journal.pone.0279425.
- Helaß, M.; Haag, G.M.; Bankstahl, U.S.; Gencer, D.; Maatouk, I. Burnout among German oncologists: a cross-sectional study in cooperation with the Arbeitsgemeinschaft Internistische Onkologie Quality of Life Working Group. J Cancer Res Clin Oncol 2023, 149(2), 765-777. doi: 10.1007/s00432-022-03937-y.
- Magdaleno, Y.; Rishel Brakey, H.; Greenberg, N.; Myers, O.; Sood, A. A qualitative review of comments by faculty who cite work-life balance as a reason to leave. Chron Mentor Coach 2022, 6(15), 587-593.
- Knox, M., Willard-Grace, R., Huang, B. et al. Maslach Burnout Inventory and a Self-Defined, Single-Item Burnout Measure Produce Different Clinician and Staff Burnout Estimates. J GEN INTERN MED 2018, 33, 1344–1351. https://doi.org/10.1007/s11606-018-4507-6.
- Hussenoeder, F.S., Bodendieck, E., Jung, F. et al. Comparing burnout and work-life balance among specialists in internal medicine: the role of inpatient vs. outpatient workplace. J Occup Med Toxicol 2021, 16, 5. https://doi.org/10.1186/s12995-021-00294-3.
- Howe, A. Doctors’ health and wellbeing. BMJ 2013; 347, https://doi.org/10.1136/bmj.f5558.
3.
“As a more quantitative remark or comment is referred to on page 8, where there is no explanation of the data at all but a Table itself the only information on this page. The same problem occurs on pages 4-7, but not in an extensive way.”
We explained the data given in tables and figures, e.g.:
The results of a significant model of the multivariate analysis of variance showed a significantly higher level of life satisfaction in the case of good assessment of economic status and among younger physicians (Figure 2).
Good economic status was significantly important for a higher quality of life in both age groups (Figure 2).
In younger physicians, low BMI and practicing sports had an additional signifi-cant influence, while in older physicians, working in the hospital and not taking sick leave were significant (Figure 2).
Reviewer 2 Report
Please, see attached file.

Author Response
Thank you very much for detailed review of our manuscript and all the comments. Our replies and changes in the manuscript are written in a red.
1.
“Abstract: Lines 15- The lack of current studies prompted us to evaluate life satisfaction of the physicians in the Silesian Province in relation to the selected factors. selected factors?”
We completed the aim of the study:
The lack of current studies prompted us to evaluate life satisfaction of the physicians in the Silesian Province in relation to the selected factors (health status, professional preferences, family and material status).
2.
“Abstract: Please choose whether or not to structure the abstract with subsections (background, methods, results, and conclusion). This version is mixed.”
We added headlines: background, material and methods, results, and conclusions
3.
“When summarizing the results, only reference to physicians is made: what about the dentists?”
We added in the text: physicians and dentists.
4.
“Abstract: Lines 32-34: How are the results of the study relevant to these professions? What contribution do they or could they make (in terms of future direction)?”
We rearranged the conclusions in the abstract:
Due to its association with the profession, the mean level of life satisfaction among the physicians and dentists prompts verification of crucial spheres related to the physical, emotional, social and material well-being and the activity of the professional group.
5.
“Introduction: Sources are missing throughout the introductory section (except for a few given in the middle paragraphs). Please add the sources on which you relied.”
We added sources: [1-5], [9,10]
- Bradfield, O.; Jenkins, K.; Spittal, M.; Bismark, M. Australian and New Zealand doctors' experiences of disciplinary notifications, investigations, proceedings and interventions relating to alleged mental health impairment: a qualitative analysis of interviews. Int J Law Psychiatry 2023, 86:101857. doi: 10.1016/j.ijlp.2022.101857.
- Lele, K.; McLean, L.M.; Peisah, C. Beyond burnout I: Doctors health services and unmet need. Australas Psychiatry 2023:10398562231159977. doi: 10.1177/10398562231159977.
- Nguyen, O.T.; Turner, K.; Parekh, A.; Alishahi Tabriz, A.; Hanna, K.; Merlo, L.J.; Hong, Y.R. Merit-based incentive payment system participation and after-hours documentation among US office-based physicians: Findings from the 2021 National Electronic Health Records Survey. J Eval Clin Pract 2023, 29(2), 397-402. doi: 10.1111/jep.13796.
- Kartsonaki, M.G.; Georgopoulos, D.; Kondili, E.; Nieri, A.S.; Alevizaki, A.; Nyktari, V.; Papaioannou, A. Prevalence and factors associated with compassion fatigue, compassion satisfaction, burnout in health professionals. Nurs Crit Care 2023, 28(2), 225-235. doi: 10.1111/nicc.12769.
- Kase, J.; Doolittle, B. Job and life satisfaction among emergency physicians: A qualitative study. PLoS One 2023, 18(2): doi: 10.1371/journal.pone.0279425.
- Helaß, M.; Haag, G.M.; Bankstahl, U.S.; Gencer, D.; Maatouk, I. Burnout among German oncologists: a cross-sectional study in cooperation with the Arbeitsgemeinschaft Internistische Onkologie Quality of Life Working Group. J Cancer Res Clin Oncol 2023, 149(2), 765-777. doi: 10.1007/s00432-022-03937-y.
- Magdaleno, Y.; Rishel Brakey, H.; Greenberg, N.; Myers, O.; Sood, A. A qualitative review of comments by faculty who cite work-life balance as a reason to leave. Chron Mentor Coach 2022, 6(15), 587-593.
6.
“Lines 81-84: Is the situation different for dentists? Since the study refers to both professional groups, the similarities and differences must be highlighted. Otherwise, it must be explained why the two professional groups are considered as one.”
We explained in the text in section Material and methods:
Dentists constituted less than 1% (n=6), therefore they were not distinguished as a separate group.
7.
Material and Methods: There is a lot of information in this section. After describing the survey and background, I suggest using subsections to better organize the information provided: Participants (all the information about the sample should be collected and presented here). Measures (describe each instrument: the aim, the source, the Likert response scale, an example item, the Cronbach alpha found in the present study). Data analysis (lines 123-142).
We did as suggested by the reviewer: We added headlines of subsections (Participants, Methods, Ethics, Funding, Data analysis).
We describe precisely Methods:
Methods
The study was conducted from January to December 2018 using a paper ques-tionnaire (Paper and Pencil Interview - PAPI - method) in the form of an anonymous survey containing closed and open questions (standard information on sociodemo-graphic, economic, occupational, lifestyle and health data).
Life satisfaction was assessed according to SWLS scale (Satisfaction with Life Scale) [13], the presence of anxiety and/or depressive symptoms - according to the Hospital Anxiety and Depression Scale (HADS) [14]. Moreover, the level of profes-sional satisfaction was evaluated [15].
SWLS contains five items (e.g. “I am satisfied with my life”, “I have achieved the most important things I wanted in my life”). Using a 7-point Likert scale, the respond-ent assesses to what extent each of the statements applies to his/her life. The total number of points range from 5 to 35 points [13]. The SWLS scale was obtained from the Laboratory of Psychological Tests of the Polish Psychological Association.
HADS is a self-report questionnaire. Patients are asked to choose one answer out of four possible for each question. Anxiety questions are marked as "A" (7 questions) and depression questions as "D" (7 questions). They are arranged alternately. The scores (from 0 to 3 points) for each question for "A" and separately for "D" are added together to obtain two scores: depression (D) and anxiety (A). A total score of 0 to 7 points indicates no disorder, values from 8 to 10 points are borderline, and ≥11 points indicate the presence of anxiety (A) and/or depressive symptoms (D) [14].
Professional satisfaction was measured according to the Canadian questionnaire validated for Polish conditions by Pena-Sanchez et al. [15]. The questionnaire consists of 17 questions evaluating various aspects of the doctor's work (e.g. relationship with other doctors, doctor-patient relationship, income, professional development opportu-nities, opportunities to separate professional duties from personal life). It uses a 6-point scale (very dissatisfied, dissatisfied, rather dissatisfied, somewhat satisfied, satisfied, very satisfied). The total number of points range from 17 to 102 points
8.
“The theoretical framework used to conduct the study should be prefaced (at the end of the Introduction or in the Material and Methods sections), as it provides explanations for the selection of variables used in the study.”
We did it at the end of Introduction:
According to the literature, there is a lack of current research on the assessment of life satisfaction among the physicians in the Silesian Province, Poland. Furthermore, little data are available on their health status, including mental health, professional preferences, family and material status. Therefore, the above aspects were addressed in this study. The model of the quality of life developed by Fence and Perry was the basis of the analysis. Introduced in 1996, the model includes the following crucial spheres and criteria describing the quality of life: physical well-being (health, physical fitness and safety), emotional well-being (mental status and self-esteem), social well-being (family and social relationships), material well-being (financial status, possession) and development and activity (active participation in life) [11].
So, the aim of the study was to evaluate the life satisfaction among Silesian physicians and to analyse its association with demographic, anthropometric and socioeconomic data, moreover with professional satisfaction, work environment, lifestyle and health status including depressive and anxiety symptoms.
9.
“The discussion section should be introduced with a brief summary of the objective and results of the study.”
We added a brief summary of results at the beginning of the discussion:
The results reflect life satisfaction among the physicians and dentists in the Sile-sian Province, Poland, in relation to the selected demographic, medical, psychological, socioeconomic and environmental factors. As shown in the study, life satisfaction among the physicians from the Silesian Province, Poland, was at an average level. Age and economic status were significant predictors. Additionally, significant predictors included BMI and practicing sports in the younger subjects (25-50 years). In turn, in the older subjects (50-80 years), these predictors were related to hospital work and be-ing on sick leave. The study found a moderate association between life satisfaction and professional satisfaction. Additionally, lower levels were reported in the subjects who presented with anxiety and/or depressive symptoms.
10.
“Then the results should be discussed comparing them with the literature. As written, it seems to be more of a literature review. Also, too many details are given about the results of other studies: it would be better to describe them in terms of "high and low levels" for example, rather than giving each percentage.”
We rearranged the discussion, added some new text, and added more references. We removed the exact results from the literature discussion, as it was suggested.
11.
“There is a reference to burnout (lines 263-272): Since this topic was not explored in the study, this comment should be moved to the end of the discussion, as a focus for future studies on the same occupational groups.”
We moved the text about burnout to the end.
12.
“Conclusions: This section should highlight the relevance of the study and how the results could be useful.”
We rearranged the conclusions:
Conclussions:
Due to its association with the profession, the mean level of life satisfaction among the physicians and dentists prompt verification of crucial spheres related to the physical, emotional, social and economic well-being and the activity of this professional group.
13.
“The limitations of the study should also be pointed out here (e.g., the cross-sectional nature of the study, etc. ...). Suggestions for future developments can be made.”
We added limitations:
Limitations
Our study has some limitations. Firstly, it was a cross-sectional study. Secondly, the physicians and the dentists were not analyzed separately because only a few den-tists (n=6) returned the questionnaire and they ultimately accounted for less than 1% of all respondents. Thirdly, we do not analyze occupational burnout and its effect on life satisfaction in Silesian doctors. In the future, we plan to extend the research to in-clude also this parameter.
Round 2
Reviewer 2 Report
The authors have made several changes and additions. I think that the quality of the manuscript has improved throughout. I have no further comments.